# Distral: Robust Multitask Reinforcement Learning

**Yee Whye Teh, Victor Bapst, Wojciech Marian Czarnecki, John Quan,**
**James Kirkpatrick, Raia Hadsell, Nicolas Heess, Razvan Pascanu**
DeepMind
London, UK

## Abstract

Most deep reinforcement learning algorithms are data inefficient in complex and rich environments, limiting their applicability to many scenarios. One direction for improving data efficiency is multitask learning with shared neural network parameters, where efficiency may be improved through transfer across related tasks. In practice, however, this is not usually observed, because gradients from different tasks can interfere negatively, making learning unstable and sometimes even less data efficient. Another issue is the different reward schemes between tasks, which can easily lead to one task dominating the learning of a shared model. We propose a new approach for joint training of multiple tasks, which we refer to as Distral (distill & transfer learning). Instead of sharing parameters between the different workers, we propose to share a "distilled" policy that captures common behaviour across tasks. Each worker is trained to solve its own task while constrained to stay close to the shared policy, while the shared policy is trained by distillation to be the centroid of all task policies. Both aspects of the learning process are derived by optimizing a joint objective function. We show that our approach supports efficient transfer on complex 3D environments, outperforming several related methods. Moreover, the proposed learning process is more robust to hyperparameter settings and more stable—attributes that are critical in deep reinforcement learning.

## 1 Introduction

Deep Reinforcement Learning is an emerging subfield of Reinforcement Learning (RL) that relies on deep neural networks as function approximators that can scale RL algorithms to complex and rich environments. One key work in this direction was the introduction of DQN [21] which is able to play many games in the ATARI suite of games [1] at above human performance. However the agent requires a fairly large amount of time and data to learn effective policies and the learning process itself can be quite unstable, even with innovations introduced to improve wall clock time, data efficiency, and robustness by changing the learning algorithm [27, 33] or by improving the optimizer [20, 29]. A different approach was introduced by [12, 19, 14], whereby data efficiency is improved by training additional auxiliary tasks jointly with the RL task.

With the success of deep RL has come interest in increasingly complex tasks and a shift in focus towards scenarios in which a single agent must solve multiple related problems, either simultaneously or sequentially. Due to the large computational cost, making progress in this direction requires robust algorithms which do not rely on task-specific algorithmic design or extensive hyperparameter tuning. Intuitively, solutions to related tasks should facilitate learning since the tasks share common structure, and thus one would expect that individual tasks should require less data or achieve a higher asymptotic performance. Indeed this intuition has long been pursued in the multitask and transfer-learning literature [2, 31, 34, 5].

Somewhat counter-intuitively, however, the above is often not the result encountered in practice, particularly in the RL domain [26, 23]. Instead, the multitask and transfer learning scenarios are

frequently found to pose additional challenges to existing methods. Instead of making learning easier it is often observed that training on multiple tasks can negatively affect performances on the individual tasks, and additional techniques have to be developed to counteract this [26, 23]. It is likely that gradients from other tasks behave as noise, interfering with learning, or, in another extreme, one of the tasks might dominate the others.

In this paper we develop an approach for multitask and transfer RL that allows effective sharing of behavioral structure across tasks, giving rise to several algorithmic instantiations. In addition to some instructive illustrations on a grid world domain, we provide a detailed analysis of the resulting algorithms via comparisons to A3C [20] baselines on a variety of tasks in a first-person, visually-rich, 3D environment. We find that the Distral algorithms learn faster and achieve better asymptotic performance, are significantly more robust to hyperparameter settings, and learn more stably than multitask A3C baselines.

## 2 Distral: Distill and Transfer Learning

We propose a framework for simultaneous reinforcement learning of multiple tasks which we call Distral. Figure 1 provides a high level illustration involving four tasks. The method is founded on the notion of a shared policy (shown in the centre) which distills (in the sense of Bucila and Hinton et al. [4, 11]) common behaviours or representations from task-specific policies [26, 23]. Crucially, the distilled policy is then used to guide task-specific policies via regularization using a Kullback-Leibler (KL) divergence. The effect is akin to a shaping reward

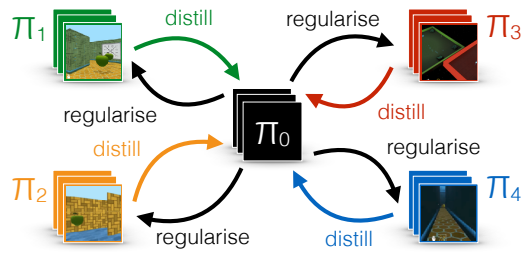

Figure 1: Illustration of the Distral framework.

which can, for instance, overcome random walk exploration bottlenecks. In this way, knowledge gained in one task is distilled into the shared policy, then transferred to other tasks.

### 2.1 Mathematical framework

In this section we describe the mathematical framework underlying Distral. A multitask RL setting is considered where there are $n$ tasks, where for simplicity we assume an infinite horizon with discount factor $\gamma$.[1] We will assume that the action $A$ and state $S$ spaces are the same across tasks; we use $a \in A$ to denote actions, $s \in S$ to denote states. The transition dynamics $p_i(s'|s, a)$ and reward functions $R_i(a, s)$ are different for each task $i$. Let $\pi_i$ be task-specific stochastic policies. The dynamics and policies give rise to joint distributions over state and action trajectories starting from some initial state, which we will also denote by $\pi_i$ by an abuse of notation.

Our mechanism for linking the policy learning across tasks is via optimising an objective which consists of expected returns and policy regularizations. We designate $\pi_0$ to be the *distilled policy* which we believe will capture agent behaviour that is common across the tasks. We regularize each task policy $\pi_i$ towards the distilled policy using $\gamma$-discounted KL divergences $\mathbb{E}_{\pi_i}[\sum_{t \geq 0} \gamma^t \log \frac{\pi_i(a_t|s_t)}{\pi_0(a_t|s_t)}]$. In addition, we also use a $\gamma$-discounted entropy regularization to further encourage exploration. The resulting objective to be maximized is:

$$J(\pi_0, \{\pi_i\}_{i=1}^n) = \sum_i \mathbb{E}_{\pi_i} \left[ \sum_{t \geq 0} \gamma^t R_i(a_t, s_t) - c_{\text{KL}} \gamma^t \log \frac{\pi_i(a_t|s_t)}{\pi_0(a_t|s_t)} - c_{\text{Ent}} \gamma^t \log \pi_i(a_t|s_t) \right]$$

$$= \sum_i \mathbb{E}_{\pi_i} \left[ \sum_{t \geq 0} \gamma^t R_i(a_t, s_t) + \frac{\gamma^t \alpha}{\beta} \log \pi_0(a_t|s_t) - \frac{\gamma^t}{\beta} \log \pi_i(a_t|s_t) \right] \quad (1)$$

where $c_{\text{KL}}, c_{\text{Ent}} \geq 0$ are scalar factors which determine the strengths of the KL and entropy regularizations, and $\alpha = c_{\text{KL}}/(c_{\text{KL}} + c_{\text{Ent}})$ and $\beta = 1/(c_{\text{KL}} + c_{\text{Ent}})$. The $\log \pi_0(a_t|s_t)$ term can be thought

of as a reward shaping term which encourages actions which have high probability under the distilled policy, while the entropy term $-\log \pi_i(a_t|s_t)$ encourages exploration. In the above we used the same regularization costs $c_{\text{KL}}, c_{\text{Ent}}$ for all tasks. It is easy to generalize to using task-specific costs; this can be important if tasks differ substantially in their reward scales and amounts of exploration needed, although it does introduce additional hyperparameters that are expensive to optimize.

## 2.2 Soft Q Learning and Distillation

A range of optimization techniques in the literature can be applied to maximize the above objective, which we will expand on below. To build up intuition for how the method operates, we will start in the simple case of a tabular representation and an alternating maximization procedure which optimizes over $\pi_i$ given $\pi_0$ and over $\pi_0$ given $\pi_i$. With $\pi_0$ fixed, (1) decomposes into separate maximization problems for each task, and is an entropy regularized expected return with redefined (regularized) reward $R_i'(a, s) := R_i(a, s) + \frac{\alpha}{\beta} \log \pi_0(a|s)$. It can be optimized using soft Q learning [10] aka G learning [7], which are based on deriving the following "softened" Bellman updates for the state and action values (see also [25, 28, 22]):

$$V_i(s_t) = \frac{1}{\beta} \log \sum_{a_t} \pi_0^\alpha(a_t|s_t) \exp\left[\beta Q_i(a_t, s_t)\right] \tag{2}$$

$$Q_i(a_t, s_t) = R_i(a_t, s_t) + \gamma \sum_{s_t} p_i(s_{t+1}|s_t, a_t) V_i(s_{t+1}) \tag{3}$$

The Bellman updates are softened in the sense that the usual max operator over actions for the state values $V_i$ is replaced by a soft-max at inverse temperature $\beta$, which hardens into a max operator as $\beta \to \infty$. The optimal policy $\pi_i$ is then a Boltzmann policy at inverse temperature $\beta$:

$$\pi_i(a_t|s_t) = \pi_0^\alpha(a_t|s_t)e^{\beta Q_i(a_t|s_t) - \beta V_i(s_t)} = \pi_0^\alpha(a_t|s_t)e^{\beta A_i(a_t|s_t)} \tag{4}$$

where $A_i(a, s) = Q_i(a, s) - V_i(s)$ is a softened advantage function. Note that the softened state values $V_i(s)$ act as the log normalizers in the above. The distilled policy $\pi_0$ can be interpreted as a policy prior, a perspective well-known in the literature on RL as probabilistic inference [32, 13, 25, 7]. However, unlike in past works, it is raised to a power of $\alpha \leq 1$. This softens the effect of the prior $\pi_0$ on $\pi_i$, and is the result of the additional entropy regularization beyond the KL divergence.

Also unlike past works, we will learn $\pi_0$ instead of hand-picking it (typically as a uniform distribution over actions). In particular, notice that the only terms in (1) depending on $\pi_0$ are:

$$\frac{\alpha}{\beta} \sum_i \mathbb{E}_{\pi_i} \left[ \sum_{t \geq 0} \gamma^t \log \pi_0(a_t|s_t) \right] \tag{5}$$

which is simply a log likelihood for fitting a model $\pi_0$ to a mixture of $\gamma$-discounted state-action distributions, one for each task $i$ under policy $\pi_i$. A maximum likelihood (ML) estimator can be derived from state-action visitation frequencies under roll-outs in each task, with the optimal ML solution given by the mixture of state-conditional action distributions. Alternatively, in the non-tabular case, stochastic gradient ascent can be employed, which leads precisely to an update which distills the task policies $\pi_i$ into $\pi_0$ [4, 11, 26, 23]. Note however that in our case the distillation step is derived naturally from a KL regularized objective on the policies. Another difference from [26, 23] and from prior works on the use of distillation in deep learning [4, 11] is that the distilled policy is "fed back in" to improve the task policies when they are next optimized, and serves as a conduit in which common and transferable knowledge is shared across the task policies.

It is worthwhile here to take pause and ponder the effect of the extra entropy regularization. First suppose that there is no extra entropy regularisation, $\alpha = 1$, and consider the simple scenario of only $n = 1$ task.Then (5) is maximized when the distilled policy $\pi_0$ and the task policy $\pi_1$ are equal, and the KL regularization term is 0. Thus the objective reduces to an unregularized expected return, and so the task policy $\pi_1$ converges to a greedy policy which locally maximizes expected returns. Another way to view this line of reasoning is that the alternating maximization scheme is equivalent to trust-region methods like natural gradient or TRPO [24, 29] which use a KL ball centred at the previous policy, and which are understood to converge to greedy policies.

If $\alpha < 1$, there is an additional entropy term in (1). So even with $\pi_0 = \pi_1$ and $\text{KL}(\pi_1\|\pi_0) = 0$, the objective (1) will no longer be maximized by greedy policies. Instead (1) reduces to an entropy

regularized expected returns with entropy regularization factor $\beta' = \beta/(1-\alpha) = 1/c_{\text{Ent}}$, so that the optimal policy is of the Boltzmann form with inverse temperature $\beta'$ [25, 7, 28, 22]. In conclusion, by including the extra entropy term, we can guarantee that the task policy will not turn greedy, and we can control the amount of exploration by adjusting $c_{\text{Ent}}$ appropriately.

This additional control over the amount of exploration is essential when there are more than one task. To see this, imagine a scenario where one of the tasks is easier and is solved first, while other tasks are harder with much sparser rewards. Without the entropy term, and before rewards in other tasks are encountered, both the distilled policy and all the task policies can converge to the one that solves the easy task. Further, because this policy is greedy, it can insufficiently explore the other tasks to even encounter rewards, leading to sub-optimal behaviour. For single-task RL, the use of entropy regularization was recently popularized by Mnih et al. [20] to counter premature convergence to greedy policies, which can be particularly severe when doing policy gradient learning. This carries over to our multitask scenario as well, and is the reason for the additional entropy regularization.

## 2.3 Policy Gradient and a Better Parameterization

The above method alternates between maximization of the distilled policy $\pi_0$ and the task policies $\pi_i$, and is reminiscent of the EM algorithm [6] for learning latent variable models, with $\pi_0$ playing the role of parameters, while $\pi_i$ plays the role of the posterior distributions for the latent variables. Going beyond the tabular case, when both $\pi_0$ and $\pi_i$ are parameterized by, say, deep networks, such an alternating maximization procedure can be slower than simply optimizing (1) with respect to task and distilled policies jointly by stochastic gradient ascent. In this case the gradient update for $\pi_i$ is simply given by policy gradient with an entropic regularization [20, 28], and can be carried out within a framework like advantage actor-critic [20].

A simple parameterization of policies would be to use a separate network for each task policy $\pi_i$, and another one for the distilled policy $\pi_0$. An alternative parameterization, which we argue can result in faster transfer, can be obtained by considering the form of the optimal Boltzmann policy (4). Specifically, consider parameterising the distilled policy using a network with parameters $\theta_0$,

$$\hat{\pi}_0(a_t|s_t) = \frac{\exp(h_{\theta_0}(a_t|s_t))}{\sum_{a'} \exp(h_{\theta_0}(a'|s_t))} \tag{6}$$

and estimating the soft advantages[2] using another network with parameters $\theta_i$:

$$\hat{A}_i(a_t|s_t) = f_{\theta_i}(a_t|s_t) - \frac{1}{\beta} \log \sum_a \hat{\pi}_0^\alpha(a|s_t) \exp(\beta f_{\theta_i}(a|s_t)) \tag{7}$$

We used hat notation to denote parameterized approximators of the corresponding quantities. The policy for task $i$ then becomes parameterized as,

$$\hat{\pi}_i(a_t|s_t) = \hat{\pi}_0^\alpha(a_t|s_t) \exp(\beta \hat{A}_i(a_t|s_t)) = \frac{\exp(\alpha h_{\theta_0}(a_t|s_t) + \beta f_{\theta_i}(a_t|s_t))}{\sum_{a'} \exp((\alpha h_{\theta_0}(a'|s_t) + \beta f_{\theta_i}(a'|s_t))} \tag{8}$$

This can be seen as a two-column architecture for the policy, with one column being the distilled policy, and the other being the adjustment required to specialize to task $i$.

Given the parameterization above, we can now derive the policy gradients. The gradient wrt to the task specific parameters $\theta_i$ is given by the standard policy gradient theorem [30],

$$\nabla_{\theta_i} J = \mathbb{E}_{\hat{\pi}_i} \left[ \left( \sum_{t \geq 1} \nabla_{\theta_i} \log \hat{\pi}_i(a_t|s_t) \right) \left( \sum_{u \geq 1} \gamma^u (R_i^{\text{reg}}(a_u, s_u)) \right) \right]$$

$$= \mathbb{E}_{\hat{\pi}_i} \left[ \sum_{t \geq 1} \nabla_{\theta_i} \log \hat{\pi}_i(a_t|s_t) \left( \sum_{u \geq t} \gamma^u (R_i^{\text{reg}}(a_u, s_u)) \right) \right] \tag{9}$$

where $R_i^{\text{reg}}(a, s) = R_i(a, s) + \frac{\alpha}{\beta} \log \hat{\pi}_0(a|s) - \frac{1}{\beta} \log \hat{\pi}_i(a|s)$ is the regularized reward. Note that the partial derivative of the entropy in the integrand has expectation $\mathbb{E}_{\hat{\pi}_i}[\nabla_{\theta_i} \log \hat{\pi}_i(a_t|s_t)] = 0$ because of the log-derivative trick. If a value baseline is estimated, it can be subtracted from the regularized

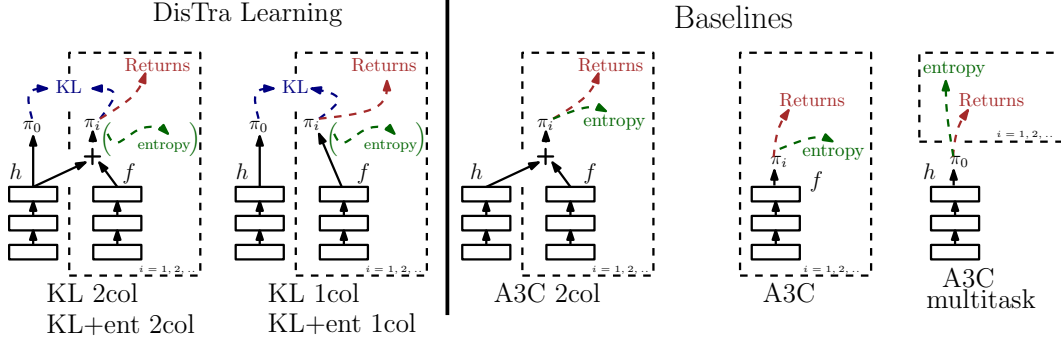

Figure 2: Depiction of the different algorithms and baselines. On the left are two of the Distral algorithms and on the right are the three A3C baselines. Entropy is drawn in brackets as it is optional and only used for KL+ent 2col and KL+ent 1col.

returns as a control variate. The gradient wrt $\theta_0$ is more interesting:

$$\nabla_{\theta_0} J = \sum_i \mathbb{E}_{\hat{\pi}_i} \left[ \sum_{t \geq 1} \nabla_{\theta_0} \log \hat{\pi}_i(a_t | s_t) \left( \sum_{u \geq t} \gamma^u (R_i^{\text{reg}}(a_u, s_u)) \right) \right] \qquad (10)$$

$$+ \frac{\alpha}{\beta} \sum_i \mathbb{E}_{\hat{\pi}_i} \left[ \sum_{t \geq 1} \gamma^t \sum_{a_t'} (\hat{\pi}_i(a_t' | s_t) - \hat{\pi}_0(a_t' | s_t)) \nabla_{\theta_0} h_{\theta_0}(a_t' | s_t) \right]$$

Note that the first term is the same as for the policy gradient of $\theta_i$. The second term tries to match the probabilities under the task policy $\hat{\pi}_i$ and under the distilled policy $\hat{\pi}_0$. The second term would not be present if we simply parameterized $\pi_i$ using the same architecture $\hat{\pi}_i$, but do not use a KL regularization for the policy. The presence of the KL regularization gets the distilled policy to learn to be the centroid of all task policies, in the sense that the second term would be zero if $\hat{\pi}_0(a_t' | s_t) = \frac{1}{n} \sum_i \hat{\pi}_i(a_t' | s_t)$, and helps to transfer information quickly across tasks and to new tasks.

## 2.4 Other Related Works

The centroid and star-shaped structure of Distral is reminiscent of ADMM [3], elastic-averaging SGD [35] and hierarchical Bayes [9]. Though a crucial difference is that while ADMM, EASGD and hierarchical Bayes operate in the space of parameters, in Distral the distilled policy learns to be the centroid in the space of policies. We argue that this is semantically more meaningful, and may contribute to the observed robustness of Distral by stabilizing learning. In our experiments we find indeed that absence of the KL regularization significantly affects the stability of the algorithm.

Another related line of work is guided policy search [17, 18, 15, 16]. These focus on single tasks, and uses trajectory optimization (corresponding to task policies here) to guide the learning of a policy (corresponding to the distilled policy $\pi_0$ here). This contrasts with Distral, which is a multitask setting, where a learnt $\pi_0$ is used to facilitate transfer by sharing common task-agnostic behaviours, and the main outcome of the approach are instead the task policies.

Our approach is also reminiscent of recent work on option learning [8], but with a few important differences. We focus on using deep neural networks as flexible function approximators, and applied our method to rich 3D visual environments, while Fox et al. [8] considered only the tabular case. We argue for the importance of an additional entropy regularization besides the KL regularization. This lead to an interesting twist in the mathematical framework allowing us to separately control the amounts of transfer and of exploration. On the other hand Fox et al. [8] focused on the interesting problem of learning multiple options (distilled policies here). Their approach treats the assignment of tasks to options as a clustering problem, which is not easily extended beyond the tabular case.

## 3 Algorithms

The framework we just described allows for a number of possible algorithmic instantiations, arising as combinations of objectives, algorithms and architectures, which we describe below and summarize in Table 1 and Figure 2. *KL divergence vs entropy regularization*: With $\alpha = 0$, we get a purely

|  | $h_{\theta_0}(a\|s)$ | $f_{\theta_i}(a\|s)$ | $\alpha h_{\theta_0}(a\|s) + \beta f_{\theta_i}(a\|s)$ |
|---|---|---|---|
| $\alpha = 0$ | A3C multitask | A3C | A3C 2col |
| $\alpha = 1$ |  | KL 1col | KL 2col |
| $0 < \alpha < 1$ |  | KL+ent 1col | KL+ent 2col |

Table 1: The seven different algorithms evaluated in our experiments. Each column describes a different architecture, with the column headings indicating the logits for the task policies. The rows define the relative amount of KL vs entropy regularization loss, with the first row comprising the A3C baselines (no KL loss).

entropy-regularized objective which does not couple and transfer across tasks [20, 28]. With $\alpha = 1$, we get a purely KL regularized objective, which does couple and transfer across tasks, but might prematurely stop exploration if the distilled and task policies become similar and greedy. With $0 < \alpha < 1$ we get both terms. *Alternating vs joint optimization*: We have the option of jointly optimizing both the distilled policy and the task policies, or optimizing one while keeping the other fixed. Alternating optimization leads to algorithms that resemble policy distillation/actor-mimic [23, 26], but are iterative in nature with the distilled policy feeding back into task policy optimization. Also, soft Q learning can be applied to each task, instead of policy gradients. While alternating optimization can be slower, evidence from policy distillation/actor-mimic indicate it might learn more stably, particularly for tasks which differ significantly. *Separate vs two-column parameterization*: Finally, the task policy can be parameterized to use the distilled policy (8) or not. If using the distilled policy, behaviour distilled into the distilled policy is "immediately available" to the task policies so transfer can be faster. However if the process of transfer occurs too quickly, it might interfere with effective exploration of individual tasks.

From this spectrum of possibilities we consider four concrete instances which differ in the underlying network architecture and distillation loss, identified in Table 1. In addition, we compare against three A3C baselines. In initial experiments we explored two variants of A3C: the original method [20] and the variant of Schulman et al. [28] which uses entropy regularized returns. We did not find significant differences for the two variants in our setting, and chose to report only the original A3C results for clarity in Section 4. Further algorithmic details are provided in the Appendix.

## 4 Experiments

We demonstrate the various algorithms derived from our framework, firstly using alternating optimization with soft Q learning and policy distillation on a set of simple grid world tasks. Then all seven algorithms will be evaluated on three sets of challenging RL tasks in partially observable 3D environments.

### 4.1 Two room grid world

To give better intuition for the role of the distilled behaviour policy, we considered a set of tasks in a grid world domain with two rooms connected by a corridor (see Figure 3) [8]. Each task is distinguished by a different randomly chosen goal location and each MDP state consists of the map location, the previous action and the previous reward. A Distral agent is trained using only the KL regularization and an optimization algorithm which alternates between soft Q learning and policy distillation. Each soft Q learning iteration learns using a rollout of length 10.

To determine the benefit of the distilled policy, we compared the Distral agent to one which soft Q learns a separate policy for each task. The learning curves are shown in Figure 3 (left). We see that the Distral agent is able to learn significantly faster than single-task agents. Figure 3 (right) visualizes the distilled policy (probability of next action given position and previous action), demonstrating that the agent has learnt a policy which guides the agent to move consistently in the same direction through the corridor in order to reach the other room. This allows the agent to reach the other room faster and helps exploration, if the agent is shown new test tasks. In Fox et al. [8] two separate options are learnt, while here we learn a single distilled policy which conditions on more past information (previous action and reward).

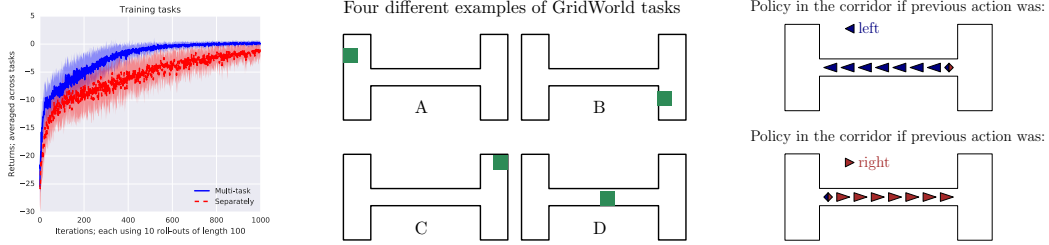

Figure 3: **Left:** Learning curves on two room grid world. The Distral agent (blue) learns faster, converges towards better policies, and demonstrates more stable learning overall. **Center:** Example of tasks. Green is goal position which is uniformly sampled for each task. Starting position is uniformly sampled at the beginning of each episode. **Right:** depiction of learned distilled policy $\pi_0$ only in the corridor, conditioned on previous action being left/right and no previous reward. Sizes of arrows depict probabilities of actions. Note that up/down actions have negligible probabilities. The model learns to preserve direction of travel in the corridor.

## 4.2 Complex Tasks

To assess Distral under more challenging conditions, we use a complex first-person partially observed 3D environment with a variety of visually-rich RL tasks. All agents were implemented with a distributed Python/TensorFlow code base, using 32 workers for each task and learnt using asynchronous RMSProp. The network columns contain convolutional layers and an LSTM and are uniform across experiments and algorithms. We tried three values for the entropy costs $\beta$ and three learning rates $\epsilon$. Four runs for each hyperparameter setting were used. All other hyperparameters were fixed to the single-task A3C defaults and, for the `KL+ent 1col` and `KL+ent 2col` algorithms, $\alpha$ was fixed at 0.5.

**Mazes** In the first experiment, each of $n = 8$ tasks is a different maze containing randomly placed rewards and a goal object. Figure 4.A1 shows the learning curves for all seven algorithms. Each curve is produced by averaging over all 4 runs and 8 tasks, and selecting the best settings for $\beta$ and $\epsilon$ (as measured by the area under the learning curves). The Distral algorithms learn faster and achieve better final performance than all three A3C baselines. The two-column algorithms learn faster than the corresponding single column ones. The Distral algorithms without entropy learn faster but achieve lower final scores than those with entropy, which we believe is due to insufficient exploration towards the end of learning.

We found that both multitask A3C and two-column A3C can learn well on some runs, but are generally unstable—some runs did not learn well, while others may learn initially then suffer degradation later. We believe this is due to negative interference across tasks, which does not happen for Distral algorithms. The stability of Distral algorithms also increases their robustness to hyperparameter selection. Figure 4.A2 shows the final achieved average returns for all 36 runs for each algorithm, sorted in decreasing order. We see that Distral algorithms have a significantly higher proportion of runs achieving good returns, with `KL+ent_2col` being the most robust.

Distral algorithms, along with multitask A3C, use a distilled or common policy which can be applied on all tasks. Panels B1 and B2 in Figure 4 summarize the performances of the distilled policies. Algorithms that use two columns (`KL_2col` and `KL+ent_2col`) obtain the best performance, because policy gradients are also directly propagated through the distilled policy in those cases. Moreover, panel B2 reveals that Distral algorithms exhibit greater stability as compared to traditional multitask A3C. We also observe that `KL` algorithms have better-performing distilled policies than `KL+ent` ones. We believe this is because the additional entropy regularisation allows task policies to diverge more substantially from the distilled policy. This suggests that annealing the entropy term or increasing the KL term throughout training could improve the distilled policy performance, if that is of interest.

**Navigation** We experimented with $n = 4$ navigation and memory tasks. In contrast to the previous experiment, these tasks use random maps which are procedurally generated on every episode. The first task features reward objects which are randomly placed in a maze, and the second task requires to return these objects to the agent's start position. The third task has a single goal object which must be repeatedly found from different start positions, and on the fourth task doors are randomly opened and

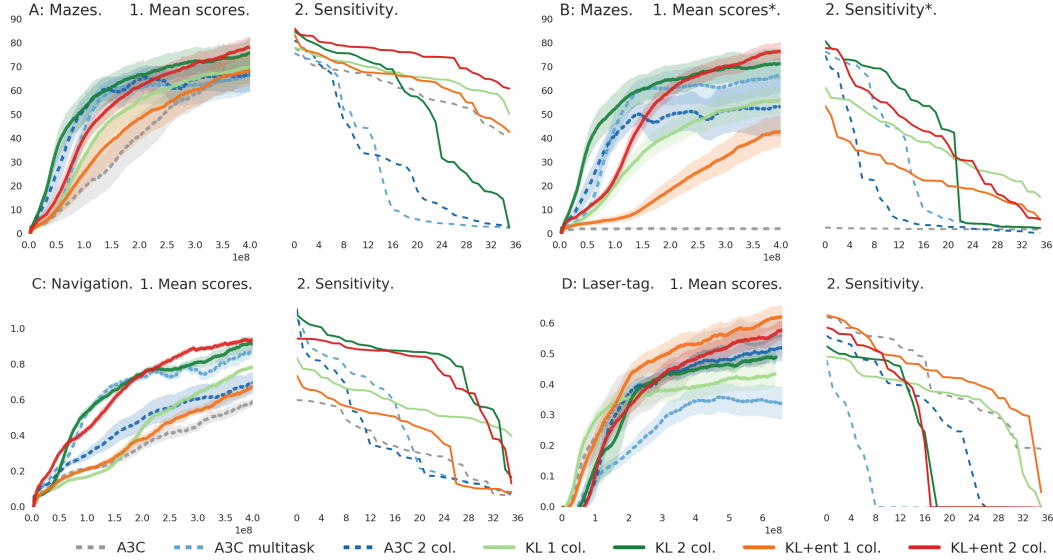

Figure 4: Panels A1, C1, D1 show task specific policy performance (averaged across all the tasks) for the maze, navigation and laser-tag tasks, respectively. The $x$-axes are total numbers of training environment steps per task. Panel B1 shows the mean scores obtained with the distilled policies (`A3C` has no distilled policy, so it is represented by the performance of an untrained network.). For each algorithm, results for the best set of hyperparameters (based on the area under curve) are reported. The bold line is the average over 4 runs, and the colored area the average standard deviation over the tasks. Panels A2, B2, C2, D2 shows the corresponding final performances for the 36 runs of each algorithm ordered by best to worst (9 hyperparameter settings and 4 runs).

closed to force novel path-finding. Hence, these tasks are more involved than the previous navigation tasks. The panels C1 and C2 of Figure 4 summarize the results. We observe again that Distral algorithms yield better final results while having greater stability (Figure 4.C2). The top-performing algorithms are, again, the 2 column Distral algorithms (`KL_2col` and `KL+ent_2col`).

**Laser-tag** In the final set of experiments, we use $n = 8$ laser-tag levels. These tasks require the agent to learn to tag bots controlled by a built-in AI, and differ substantially: fixed versus procedurally generated maps, fixed versus procedural bots, and complexity of agent behaviour (e.g. learning to jump in some tasks). Corresponding to this greater diversity, we observe (see panels D1 and D2 of Figure 4) that the best baseline is the A3C algorithm that is trained independently on each task. Among the Distral algorithms, the single column variants perform better, especially initially, as they are able to learn task-specific features separately. We observe again the early plateauing phenomenon for algorithms that do not possess an additional entropy term. While not significantly better than the A3C baseline on these tasks, the Distral algorithms clearly outperform the multitask A3C.

**Discussion** Considering the 3 different sets of complex 3D experiments, we argue that the Distral algorithms are promising solutions to the multitask deep RL problem. Distral can perform significantly better than A3C baselines when tasks have sufficient commonalities for transfer (maze and navigation), while still being competitive with A3C when there is less transfer possible. In terms of specific algorithmic proposals, the additional entropy regularization is important in encouraging continued exploration, while two column architectures generally allow faster transfer (but can affect performance when there is little transfer due to task interference). The computational costs of Distral algorithms are at most twice that of the corresponding A3C algorithms, as each agent need to process two network columns instead of one. However in practice the runtimes are just slightly more than for A3C, because the cost of simulating environments is significant and the same whether single or multitask.

# 5 Conclusion

We have proposed Distral, a general framework for distilling and transferring common behaviours in multitask reinforcement learning. In experiments we showed that the resulting algorithms learn quicker, produce better final performances, and are more stable and robust to hyperparameter settings. We have found that Distral significantly outperforms the standard way of using shared neural network parameters for multitask or transfer reinforcement learning.

Two ideas in Distral might be worth reemphasizing here. We observe that distillation arises naturally as one half of an optimization procedure when using KL divergences to regularize the output of task models towards a distilled model. The other half corresponds to using the distilled model as a regularizer for training the task models. Another observation is that parameters in deep networks do not typically by themselves have any semantic meaning, so instead of regularizing networks in parameter space, it is worthwhile considering regularizing networks in a more semantically meaningful space, e.g. of policies.

We would like to end with a discussion of the various difficulties faced by multitask RL methods. The first is that of positive transfer: when there are commonalities across tasks, how does the method achieve this transfer and lead to better learning speed and better performance on new tasks in the same family? The core aim of Distral is this, where the commonalities are exhibited in terms of shared common behaviours. The second is that of task interference, where the differences among tasks adversely affect agent performance by interfering with exploration and the optimization of network parameters. This is the core aim of the policy distillation and mimic works [26, 23]. As in these works, Distral also learns a distilled policy. But this is further used to regularise the task policies to facilitate transfer. This means that Distral algorithms can be affected by task interference. It would be interesting to explore ways to allow Distral (or other methods) to automatically balance between increasing task transfer and reducing task interference.

Other possible directions of future research include: combining Distral with techniques which use auxiliary losses [12, 19, 14], exploring use of multiple distilled policies or latent variables in the distilled policy to allow for more diversity of behaviours, exploring settings for continual learning where tasks are encountered sequentially, and exploring ways to adaptively adjust the KL and entropy costs to better control the amounts of transfer and exploration. Finally, theoretical analyses of Distral and other KL regularization frameworks for deep RL would help better our understanding of these recent methods.

## Footnotes

[1]The method can be easily generalized to other scenarios like undiscounted finite horizon.

[2]In practice, we do not actually use these as advantage estimates. Instead we use (8) to parameterize a policy which is optimized by policy gradients.

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
