[Supplementary Material]

# Distral: Robust Multitask Reinforcement Learning —Supplementary Materials—

**Yee Whye Teh, Victor Bapst, Wojciech Marian Czarnecki, John Quan,**
**James Kirkpatrick, Raia Hadsell, Nicolas Heess, Razvan Pascanu**
DeepMind
London, UK

## A    Algorithms

A description of all the algorithms tested:

- `A3C`: A policy is trained separately for each task using A3C.
- `A3C_multitask`: A single policy is trained using A3C by simultaneously training on all tasks.
- `A3C_2col`: A policy is trained for each task using A3C, which is parameterized using a two-column architecture with one column shared across tasks (8).
- `KL_1col`: Each policy including the distilled policy is parameterized by one network, and trained to optimize (1) with $\alpha = 1$, i.e. only using the KL regularization.
- `KL+ent_1col`: Same as `KL_1col` but using both KL and entropy regularization. We set $\alpha = 0.5$ and did not tune for it in our experiments.
- `KL_2col`: Same as `KL_1col` but using the two-column architecture (8) with one shared network column which also produces the distilled policy $\hat{\pi}_0$.
- `KL+ent_2col`: Same as `KL+ent_1col` but using the two-column architecture.

## B    Experimental details

### B.1    Two room grid world

The agent can stay put or move in each of the four cardinal coordinates. A penalty of $-0.1$ is incurred for every time step and a penalty of $-0.5$ is incurred if the agent runs into the wall. On reaching the goal state the episode terminates and the agent gets a reward of $+1$. We used learning rate of $0.1$, discount of $0.95$ for soft Q learning, $\beta = 5$, and regularized the distilled policy by using a pseudocount of 1 for each action in each state. The reported results are not sensitive to these settings and we did not tune for them.

### B.2    Complex 3D tasks

The three sets of tasks are described in Table 2.

We implemented the updates (9-10) by training a distributed agent *a la* A3C [20] with 32 workers for each task, coordinated using parameter servers. The agent receives a RGB observation from the environment in the form of a $3 \times 84 \times 84$ tensor. Each network column has the same architecture as in Mnih et al. [20] and consists of two convolutional layers with ReLU nonlinearities, followed by a fully connected layer with 256 hidden units and ReLU nonlinearity, which then feeds into an LSTM. Policy logits and values are then read out linearly from the LSTM. We used RMSProp as an optimizer, and batches of length 20.

We used a set of 9 hyperparameters $(1/\beta, \epsilon) \in \{3 \cdot 10^{-4}, 10^{-3}, 3 \cdot 10^{-3}\} \times \{2 \cdot 10^{-4}, 4 \cdot 10^{-4}, 8 \cdot 10^{-4}\}$ for the entropy costs and initial learning rate. We kept $\alpha = 0.5$ throughout all runs of `KL+ent_1col` and `KL+ent_2col` (and, by definition, we have $\alpha = 0$ for `A3C`, `A3C_multitask`, `A3C_2col` and $\alpha = 1$ for `KL_1col` and `KL_2col`.). The learning rate was annealed linearly from its initial value $\epsilon$ down to $\epsilon/6$ over a total of $N$ environment steps per task, where $N = 4 \cdot 10^8$ for the maze and navigation tasks, and $N = 6.6 \cdot 10^8$ for the laser-tag tasks. We used an action-repeat of 4 (each action output by the network is fed 4 times to the environment), so the number of training steps in each environment is respectively $10^8$ and $1.65 \cdot 10^8$. In addition to the updates (9-10), our implementation had small regularization terms over the state value estimates $\hat{V}_i$ for each task, in the form of $L_2$

| | Num tasks | Inter-task variability | Map/maze layout | Objectives and Agent Behaviours |
|---|---|---|---|---|
| **Mazes** | 8 | low | Fixed layout for each task. | Collect a goal object multiple times from different start positions in the maze. |
| **Navigation** | 4 | medium | Procedurally varied on every episode. | Various objectives requiring memory and exploration skills, including collecting objects and navigating back to start location, and finding a goal object from multiple start positions with doors that randomly open and close. |
| **Laser-tag** | 8 | high | Fixed for some tasks and procedurally varied for others. | Tag bots controlled by the OpenArena AI while collecting objects to increase score. May require jumping and other agent behaviours. |

Table 2: Details on the various tasks used in Section 4.2.

losses with a coefficient 0.005, to encourage a small amount of transfer of knowledge in value estimates too. We did not tune for this parameter, and believe it is not essential to our results.

Because of the small values of $\beta$ used, we parameterized $\beta$ times the soft advantages using the network outputs instead, so that (7-8) read as:

$$\beta \hat{A}_i(a_t|s_t) = f_{\theta_i}(a_t|s_t) - \log \sum_a \hat{\pi}_0^\alpha(a|s_t) \exp(f_{\theta_i}(a|s_t)) \tag{11}$$

$$\hat{\pi}_i(a_t|s_t) = \hat{\pi}_0^\alpha(a_t|s_t) \exp(\beta \hat{A}_i(a_t|s_t)) = \frac{\exp(\alpha h_{\theta_0}(a_t|s_t) + f_{\theta_i}(a_t|s_t))}{\sum_{a'} \exp((\alpha h_{\theta_0}(a'|s_t) + f_{\theta_i}(a'|s_t))} \tag{12}$$

When reporting results for the navigation and laser-tag sets of tasks, we normalized the results by the best performance of a standard A3C agent on a task by task basis, to account for different rewards scales across tasks.

# C Detailed learning curves

Figure 5: Scores on the 8 different tasks of the navigation suite. Top two rows show the results with the task specific policies, bottom two rows show the results with the distilled policy. For each algorithm, results for the best set of hyperparameters are reported, as obtained by maximizing the averaged (over tasks and runs) areas under curves. For each algorithm, the 4 thin curves correspond to the 4 runs. The average over these runs is shown in bold. The $x$-axis shows the total number of training environment steps for each task.

Figure 6: Scores on the 4 different tasks of the navigation suite. Top row shows the results with the task specific policies, bottom row shows the results with the distilled policy. For each algorithm, results for the best set of hyperparameters are reported, as obtained by maximizing the averaged (over tasks and seeds) area under curve. For each algorithm, the 4 curves correspond to the 4 different seeds. The average over these seeds is shown in bold. The $x$-axis shows the total number of training environment steps for the corresponding task.

Figure 7: Scores on the 8 different tasks of the laser-tag suite. Only results with the task policies were computed for this set of tasks. For each algorithm, results for the best set of hyperparameters are reported, as obtained by maximizing the averaged (over tasks and seeds) area under curve. For each algorithm, the 4 curves correspond to the 4 different seeds. The average over these seeds is shown in bold. The $x$-axis shows the total number of training environment steps for the corresponding task.