[Reviews · NeurIPS 2017]

Reviewer 1



Summary The described approach improves data efficiency for deep reinforcement learning through multitask learning by sharing a distilled policy between the individual task learners. Comments * Overall, most of the paper seems to be very nicely done, e.g.: ** Complete and concise description of related work and underlying reasoning/idea for the approach (section 1) ** Detailed mathematical derivation for both tabular based approaches as well as policy-gradient based models. ** Interesting connections are highlighted (e.g.reminiscence of DisTral to ADMM in section 2.3) ** Experiments seem reasonable. * I miss direct comparison(s) to the somewhat related 'Policy destillation' paper [21]. it would have been nice to see some of the same tasks used as there. The same holds also true for [18].

Reviewer 2



The paper presents an approach to performing transfer between multiple reinforcement learning tasks by regularizing the policies of different tasks towards a central policy, and also encouraging exploration in these policies. The approach relies on KL-divergence regularization. The idea is straightforward and well explained. There are no theoretical results regarding the learning speed or quality of the policies obtained (though these are soft, so clearly there would be some performance loss compared to optimal). The evaluation shows slightly better results that A3C baselines in both some simple mazes and deep net learning tasks. While the paper is well written, and the results are generally positive, the performance improvements are modest. It would have been nice also to quantify the performance in *runtime* instead of just in terms of number of samples used (since the networks architectures are different). In the tabular case, it would really have been nice to show some more detailed experiments to understand the type of behaviour obtained. Eg. what happens if goals are sampled uniformly (as in these experiments) vs all sampled in one or the other of the extreme rooms, and what happens as the number of sampled tasks increases. Also, another baseline which does not seem to be included but would be very useful is to simply regularize task policies towards literally a "centroid" of the suggested actions, without learning a distilled policy. One could simply average the policies of the different tasks and regularize towards that quantity. It is not clear if this would be competitive with the proposed approach, but conceptually it should lead to very similar results. If the number of tasks increases, one would expect the distilled policy would in fact simply become uniform - is this the case? Finally, It would really have been nice to see some attempt at theory in the tabular case. It seems to me that one ought to be able to bound the loss of this approach wrt optimal policies using a variational approach, since the optimization seems to suggest that the algorithm computes a variational approximation to what would be the optimal policy for a distribution of MDPs. Such a result would make the paper more interesting conceptually and give some confidence in its applicability beyond the domains used in the experiments. Smaller suggestions: - The title refers to "robustness" which in RL is a technical term regarding optimization wrt an unknown distribution of dynamics (using typically min-max algorithms). This is not the type of approach proposed here, so some clarification is needed.

Reviewer 3



The paper proposes an approach (DISTRAL) for the multitask learning setting. DISTRAL shares/transfers a 'distilled policy' between different tasks. The aim is that this distilled policy captures common behaviour across tasks. Experiments on several domains are performed to evaluate the approach. The topic of this paper, multitask learning within RL, is very relevant and important. Furthermore, the paper is well-written, contains clever optimization strategies and extensive experiments. The main drawback of the paper, in my opinion, is that the central learning objective (Equation 1) is not well motivated: it is not clear why this approach is a good one to follow for multitask learning. Personally, I don't see why and how it would scale to complex domains and/or domains where individual task policies are very different from each other. The experiments actually seem to confirm this: for the complex tasks of Section 4.2, the performance improvement of DISTRAL compared to the baselines (A3C/A3C 2col/A3C multitask) is minimal. What would make the paper stronger is if a careful analytical and/or empirical evaluation was done on what type of multitask domains this method does well on and on what type of domains it does not well on. Furthermore, while the underlying concept of policy distillation is interesting, it has already been published in an ICLR'16 publication (citation 21). That being said, I think the pros of this paper outweigh the cons and recommend acceptance.